# Skyrmions in synthetic antiferromagnets and their nucleation via electrical current and ultra-fast laser illumination

Roméo Juge[1,12], Naveen Sisodia[1,12], Joseba Urrestarazu Larrañaga[1], Qiang Zhang[1], Van Tuong Pham[1], Kumari Gaurav Rana[1], Brice Sarpi[2], Nicolas Mille[2], Stefan Stanescu[2], Rachid Belkhou[2], Mohamad-Assaad Mawass[3], Nina Novakovic-Marinkovic[3], Florian Kronast[3], Markus Weigand[4], Joachim Gräfe[5], Sebastian Wintz[5], Simone Finizio[6], Jörg Raabe[6], Lucia Aballe[7], Michael Foerster[7], Mohamed Belmeguenai[8], Liliana D. Buda-Prejbeanu[1], Johan Pelloux-Prayer[1], Justin M. Shaw[9], Hans T. Nembach[9,10], Laurent Ranno[11], Gilles Gaudin[1] & Olivier Boulle[1]✉

Magnetic skyrmions are topological spin textures that hold great promise as nanoscale information carriers in non-volatile memory and logic devices. While room-temperature magnetic skyrmions and their current-induced motion were recently demonstrated, the stray field resulting from their finite magnetisation and their topological charge limit their minimum size and reliable motion. Antiferromagnetic skyrmions allow to lift these limitations owing to their vanishing magnetisation and net zero topological charge, promising ultra-small and ultra-fast skyrmions. Here, we report on the observation of isolated skyrmions in compensated synthetic antiferromagnets at zero field and room temperature using X-ray magnetic microscopy. Micromagnetic simulations and an analytical model confirm the chiral antiferromagnetic nature of these skyrmions and allow the identification of the physical mechanisms controlling their size and stability. Finally, we demonstrate the nucleation of synthetic antiferromagnetic skyrmions via local current injection and ultra-fast laser excitation.

Magnetic skyrmions have raised considerable interest in the last years due to the wealth of physical phenomena they evidence, at the frontier between topology and magnetism, as well as promising applications in information technology[1–6]. Magnetic skyrmions are bi-dimensional local whirling of the magnetisation featuring a non-trivial topology.

They possess a topological charge $|Q| = 1$, i.e., the skyrmion spin texture wraps once around the unit sphere in the spin space. Their small lateral dimensions, down to a few nanometres, topological invariance and fast manipulation by electrical current can be exploited to store and manoeuvre the information at the nanoscale in non-volatile

[1]Univ. Grenoble Alpes, CNRS, CEA, SPINTEC, 38000 Grenoble, France. [2]Synchrotron SOLEIL, L'Orme des Merisiers, 91190 Saint-Aubin, France. [3]Helmholtz-Zentrum Berlin für Materialien und Energie, Albert-Einstein-Straße 15, 12489 Berlin, Germany. [4]Helmholtz-Zentrum Berlin für Materialien und Energie GmbH, Hahn-Meitner-Platz 1, D-14109 Berlin, Germany. [5]Max Planck Institute for Intelligent Systems, Heisenbergstraße 3, 70569 Stuttgart, Germany. [6]Swiss Light Source, Paul Scherrer Institut, 5232 Villigen, Switzerland. [7]ALBA Synchrotron Light Facility, 08290 Cerdanyola del Vallès, Barcelona, Spain. [8]Laboratoire des Sciences des Procedés et des Matériaux, CNRS, Univ. Paris 13, 93430 Villetaneuse, France. [9]Quantum Electromagnetics Division, National Institute of Standards and Technology, Boulder, CO 80309, USA. [10]Department of Physics, University of Colorado, Boulder, CO 80309, USA. [11]Univ. Grenoble Alpes, CNRS, Institut Néel, 38042 Grenoble, France. [12]These authors contributed equally: Roméo Juge, Naveen Sisodia. ✉e-mail: olivier.boulle@cea.fr

memory and logic devices. The recent demonstration of room-temperature skyrmions in technology-relevant ultra-thin ferromagnetic (FM) films lacking inversion symmetry as well as their fast current-induced manipulation were first important steps toward the practical realisation of such devices[7–13]. These material stacks composed of ultra-thin heavy metal/ferromagnet films combine the necessary ingredients for the stabilisation of skyrmions, namely perpendicular magnetic anisotropy (PMA), Dzyaloshinskii-Moriya interaction (DMI) and dipolar interactions. However, FM skyrmions suffer from several drawbacks limiting their implementation in functional devices. First, the non-local stray fields in FMs help stabilise skyrmions but also limit their minimal size to several tens of nanometres at room temperature[14], and an external magnetic field is generally needed for their stabilisation[10,14–16]. Second, their dynamics in tracks is impaired by the skyrmion Hall effect, a direct consequence of their finite topological charge[12,13], which deflects the skyrmions from their straight trajectory along the current, pushing them towards the edge of the device where they can be annihilated, resulting in the loss of information. Third, their non-zero topological charge further hinders the skyrmion motion by creating a topological damping[14,17], which limits their velocity to 100 m s$^{-1}$ in FMs[8,11].

These problems can be overcome by considering antiferromagnetic (AF) skyrmions, which consist of two skyrmions with antiparallel neighbouring spins, and therefore opposite topological charges, such as skyrmions in different antiferromagnetically coupled sub-lattices. AF skyrmions with tens of nm in diameter and zero-field room-temperature stability are predicted due to their vanishing magnetisation[14,15,18,19]. Furthermore, their zero net topological charge cancels the skyrmion Hall effect and simulations predict a straight trajectory along the current with velocities above 1000 m s$^{-1}$[19–23]. Finally, their zero net magnetisation makes them insensitive to external magnetic fields.

To implement AF skyrmions in devices, a first requirement is to demonstrate their room-temperature zero-field stability as well as their controlled nucleation, a challenging task given their vanishing magnetic moment. Progress has been made recently with the demonstration of topological spin textures in AFs, such as fractional AF skyrmion lattices in bulk $MnSc_2S_4$ compounds at cryogenic temperature[24] and AF half-skyrmions and bimerons in $\alpha$-$Fe_2O_3$ at room temperature[25]. However, these AF textures were detected in bulk materials or epitaxial films which are complex to grow and hardly compatible with large-scale CMOS integration. In addition, the controlled nucleation of isolated AF skyrmions at room temperature using local excitation is still lacking.

A more promising approach lies in synthetic antiferromagnets (SAFs). SAFs are composed of two (or an even number of) ultra-thin FM layers AF coupled through a non-magnetic spacer, via Ruderman–Kittel–Kasuya–Yosida (RKKY)-type interlayer exchange coupling[26,27]. SAF skyrmions thus consist of a pair (or pairs) of AF coupled skyrmions, each in its respective FM layer. In contrast to bulk AFs, the magnetic properties of ultra-thin films forming a SAF, such as PMA, DMI and interlayer exchange coupling, can be tuned by varying the nature and the thickness of the materials composing the layers, providing additional degrees of freedom to control the skyrmion size and stability. SAF skyrmions are also insensitive to moderate external magnetic fields as long as these do not outweigh the interlayer exchange coupling. Furthermore, SAFs are composed of technology-relevant sputtered films compatible with CMOS integration as well as standard spintronic devices such as magnetic tunnel junctions for information readout.

However, while skyrmions in FM can be easily nucleated from a stripe domain state using an external magnetic field, the vanishing moment of SAF skyrmions makes their nucleation and visualisation challenging. To circumvent this problem, strategies have been considered where SAF skyrmions were nucleated from a multi-domain

state using either an external field in an uncompensated SAF[28] or a bias field in a SAF exchanged-coupled to a FM biasing layer[29]. Nevertheless, these approaches have important limitations regarding current-induced skyrmion manipulation: the skyrmion Hall effect is mitigated but not suppressed in uncompensated SAFs, and the use of a biasing FM layer would lead to current shunting and a sensitivity to external magnetic fields.

Here, we demonstrate that isolated skyrmions can be stabilised in a compensated SAF at room temperature and zero external field. Using X-ray magnetic microscopy, we are able to visualise the skyrmions in the different layers constituting the SAF and to confirm their antiparallel alignment. By combining X-ray microscopy, micromagnetic simulations and analytical modelling, we confirm their internal chiral spin texture and show that their size and stability can be tailored by tuning the thickness of the constituent FM layers. Local and controlled nucleation of SAF skyrmions is also demonstrated using pulsed current injection as well as ultra-fast laser excitation. These results open up a new path for using SAF skyrmions in future logic and memory devices.

## Results and discussion

### Observation of skyrmions in SAF at room temperature

The stabilisation of SAF skyrmions requires a fine tuning of the magnetic properties of the different constituent FM layers. These must combine: (i) PMA; (ii) a large DMI so as to promote spin textures with the same chirality; (iii) a large AF coupling between the FM layers; (iv) equal magnetic moment to obtain a fully compensated SAF; (v) spin-orbit torques of the same sign on both layers, such that both AF skyrmions are moved in the same direction upon injection a current.

To combine these features, we optimised a SAF multilayer with the following composition (see Methods, SAF1): [Pt(0.5)/FM1/Ru(0.85)/Pt(0.5)/FM2/Ru(0.85)]$_2$, where FM1 = Co(0.3)/Ni$_{80}$Fe$_{20}$(1.45)/Co(0.3) and FM2 = Co(1.35) (thicknesses in nanometres) are AF coupled through the Ru spacer via interlayer exchange coupling (see Fig. 1a). The thickness of the different ferromagnets composing FM1 and FM2 is adjusted so as to reach magnetic moment compensation and, at the same time, to be close to the spin reorientation (out-of-plane to in-plane) transition (see Fig. 1b), which facilitates the skyrmion nucleation by notably reducing the domain wall (DW) energy (see Supplementary Information, S1.1.1). Here, we chose to use different FM materials in the two constituent layers. This allows us to distinguish the skyrmions in the Pt/Co/NiFe/Co and the Pt/Co AF coupled layers using transmission X-ray microscopy, by adjusting the X-ray energy to the Fe or the Co absorption edge, respectively. From Brillouin light scattering spectroscopy, we measured separately the DMI in each constituent FM layer, FM1 and FM2, to be $|D_{FM1}| = 0.67 \pm 0.04$ mJ m$^{-2}$ and $|D_{FM2}| = 0.57 \pm 0.11$ mJ m$^{-2}$, respectively (see Supplementary Information, S1.1.2). Note that the sign of the DMI extracted in these experiments is consistent with a left-handed (anticlockwise) chirality, in agreement with that expected at the Pt/Co interface[30]. In addition, the Co/Ru interface is expected to enhance the DMI by adding up to that coming from the Pt/Co interface[30–32].

Figure 1c, d displays X-ray magnetic circular dichroism scanning transmission X-ray microscopy (XMCD-STXM) images of a 2-μm-wide SAF track. The images were acquired at the Co and the Fe $L_3$ absorption edges respectively, after out-of-plane demagnetisation. The stripe domain structure is due to the low PMA of the layers. As expected, the magnetic contrast is opposite at the Co and the Fe edges, confirming that the magnetisation points in opposite directions in FM1 and FM2 and that the domains are AF coupled. A skyrmion can also be seen at the top of the image. Isolated SAF skyrmions can also be nucleated using large external magnetic fields (Fig. 1e) as well as the injection of current pulses in the track (see Supplementary Information, S1.2.3). Figure 1e shows a zero-field image of a track with stripe-like domains and an isolated skyrmion. It was obtained after applying successively an out-of-plane and an in-plane magnetic field of 180 mT. Although the

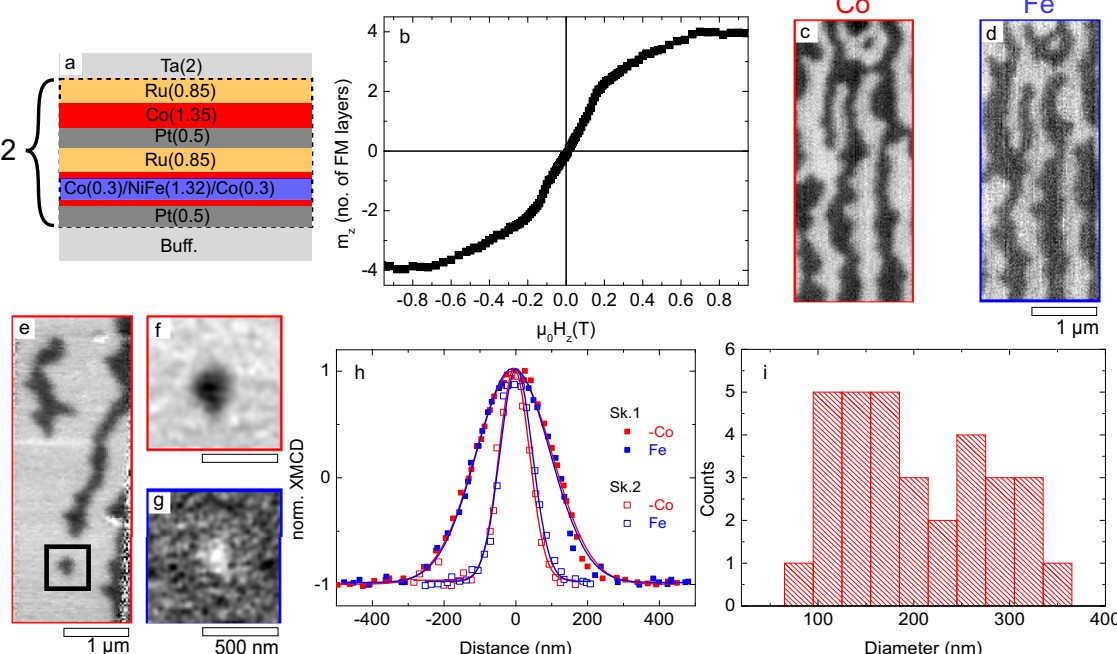

**Fig. 1 | Observation of SAF skyrmions. a** Material stack for the SAF. Buff. denotes Ta(3)/Pt(2.5) (thicknesses in nanometres). **b** Out-of-plane hysteresis loop measured by vibrating sample magnetometry. $m_z$ denotes the reduced out-of-plane component of the magnetisation. **c, d** X-ray magnetic circular dichroism scanning transmission X-ray microscopy (XMCD-STXM) images acquired at **c** the Co $L_3$ and **d** the Fe $L_3$ absorption edges after out-of-plane demagnetisation. An isolated skyrmion is observed at the top of the image (Sk.2). **e** XMCD-STXM image acquired at the Co edge showing an isolated skyrmion (Sk.1). The image was recorded at zero field after the successive application of an external magnetic field (180 mT) out of and in the plane of the layers. **f, g** XMCD microcopy images obtained from amplitudes of ptychography reconstructions of the skyrmion in the black box in **e** acquired at the Co and Fe edges, respectively. **h** Normalised XMCD signal obtained from line-scans along the diameter of skyrmions Sk.1 and Sk.2, in **c, d, f, g**, respectively. The signal for Co is inverted for comparison purposes and the solid lines are Gaussian fits. **i** Distribution of skyrmion diameters extracted from various STXM images. Scale bar in **c–e** and in **f, g** is 1 μm and 500 nm, respectively.

detailed nucleation mechanism of skyrmions in SAFs is beyond the scope of this study, one can explain the sensitivity of SAF skyrmions to external fields by the uncompensated moments in the AF coupled DWs that act as nucleation centres for magnetisation reversal in SAFs[33,34], provided that the applied field is above the RKKY field limit[29]. Figure 1f, g show higher spatial resolution images of the SAF skyrmion, obtained using ptychography reconstruction[35–40] and acquired at the Co and the Fe $L_3$ edges, respectively. The opposite magnetic contrast at the two edges demonstrates the antiparallel alignment of the skyrmion magnetisation in FM1 and FM2. To emphasise the AF coupling, the XMCD contrasts at the Co and Fe edges along the diameters of the skyrmions in Fig. 1c, d (Sk.2) and in Fig. 1f, g (Sk.1) are plotted in Fig. 1h. For comparison purpose, the Co signal is inverted and both signals are normalised. Figure 1h reveals that both signals superimpose within the resolution limit of the instrument, confirming that this magnetic texture is a SAF skyrmion. The full width at half maximum of a Gaussian fit provides a measure of the skyrmion diameter at $m_z = 0$: 107 nm and 239 nm, respectively. Figure 1i shows the distribution of skyrmion diameters obtained after repeating these nucleation procedures using either a current or a magnetic field. We find an average skyrmion diameter of 204 nm with standard deviation 85 nm. This large dispersion could be explained by inhomogeneities in the material[41–43] as well as the interaction between the skyrmions and the surrounding stripe domains in this stack. Using a local nucleation method from a single-domain state, as will be shown in the following, or by confining the skyrmions in narrower tracks could allow us to limit the skyrmion size dispersion.

## Micromagnetic simulations and analytical modelling

These experiments demonstrate stable room-temperature skyrmions in compensated SAFs at zero external magnetic field. However, they raise several questions. First, in these STXM experiments, only the out-of-plane component of the magnetisation is accessible so that the chirality of the spin texture cannot be determined. While left-handed Néel skyrmions are expected from the sign of the DMI at the Pt/Co interface[9], the multilayer structure of the SAF leads to additional competing interlayer interactions which can affect the skyrmion texture. In FM multilayers, the interlayer stray fields were shown to lead to twisted spin textures along the film thickness, with the formation of Néel caps and layer-dependent chirality[44–46]. While it is anticipated that this effect would vanish in compensated SAF multilayers due to the overall vanishing magnetic moment, the dipolar interaction between neighbouring AF coupled layers may also lead to opposite chiralities between neighbouring FM layers[17]. Second, the vanishing stray fields in SAFs are also expected to produce very small skyrmions, in the tens of nm range[47], while only large diameters are observed experimentally[29].

To address these questions, we carried out micromagnetic simulations using experimentally extracted parameters (see Supplementary Information, S1.1). Figure 2a shows a top view of the skyrmion spin texture in the middle Co layer (Co1) as well as a side view of the magnetisation across the skyrmion in the different layers at zero field. The simulations confirm the existence of the SAF skyrmion solution in our system and show that the skyrmions in the different constituent layers all exhibit the same left-handed chirality, in agreement with the sign of the DMI arising from the Pt/Co interface. The SAF skyrmions seen experimentally are hence homochiral, the chirality being driven by the DMI. This is also confirmed by additional magnetic microscopy (XMCD-PEEM) observations in similar stacks, where a direct observation of the skyrmion chirality of the top layer was achieved (see Supplementary Information, S2). Line-scans along the skyrmion diameter provide magnetisation profiles within the different layers (see Fig. 2b): these profiles overlap within the $|m_z| = 1$ regions but a slight

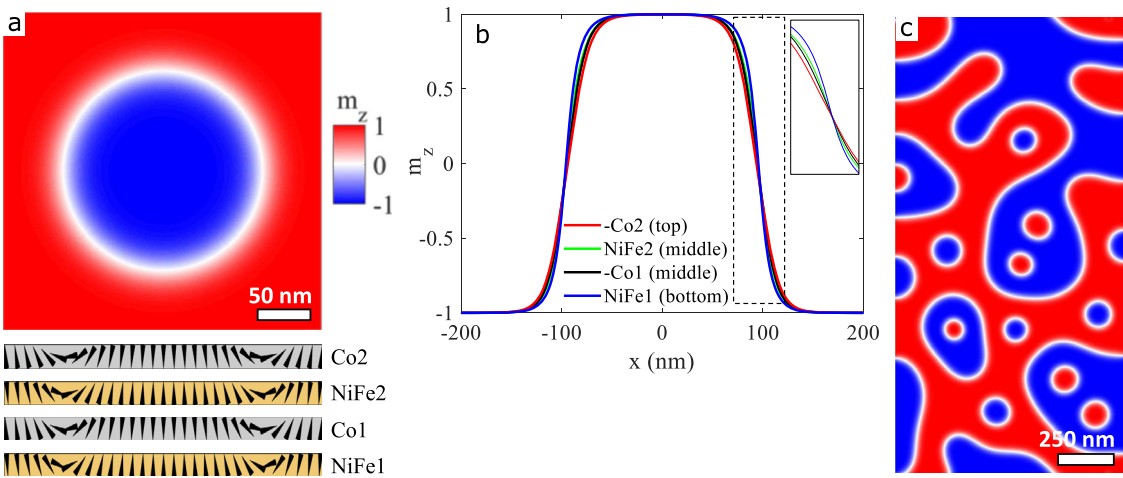

**Fig. 2 | Micromagnetic simulation of skyrmions in a SAF. a** Spin texture of a skyrmion in the Co1 layer and across the skyrmion diameter within the different FM layers. **b** Out-of-plane component of the magnetisation $m_z$ across the skyrmion diameter in the different layers. **c** Simulated spin texture in a $1 \times 2 \, \mu m^2$ strip.

misalignment is observed at the DW position. This can be accounted for by several effects related to the stray field energy: (i) a change of the DW width of the top (Co2) and bottom (NiFe1) layers induced by the uncompensated in-plane stray field component[48], and (ii) a lateral shift of the DWs[33,34]. Both these effects lead to locally uncompensated magnetic moments, which allows a decrease of the stray field energy at the expense of the interlayer exchange interaction. Additionally, layer-to-layer variation in the magnetic parameters (anisotropy and DMI) can also affect the DW width. Note that this lateral shift (less than 10 nm) could not be observed experimentally due to the limited spatial resolution of the STXM (30 nm). Figure 2c shows the domain structure on a larger scale obtained from the relaxation of an initially demagnetised state. The simulation reproduces the mixed stripe domain and skyrmion state observed experimentally.

We now discuss the large skyrmion size observed experimentally. Different energies govern the size and stability of FM and AF skyrmions in ultra-thin films[9,14,15]. On the one hand, FM skyrmions are mostly stabilised by non-local stray field energies: the skyrmion size results from a balance between the DW surface tension $\sigma \sim 4\sqrt{AK_{eff}} - \pi D$, which tends to decrease the skyrmion size, and the non-local stray field energy, which tends to increase it ($D$ is the DMI constant, $A$ is the exchange constant, $K_{eff}$ the effective perpendicular magnetic anisotropy). On the other hand, AF skyrmions with vanishing stray field energy are mostly stabilised by the DMI energy. The AF skyrmion size and stability results from a balance between (i) the DMI energy which favours larger skyrmions, (ii) the cost in anisotropy and exchange energy at larger radius, which favours smaller skyrmions, and (iii) the curvature energy cost at low radius due to the exchange energy. Eventually, the AF skyrmion size $r_0$ and stability energy $E_b$ can be approximated as[49]

$$r_0 = 1.35\Delta \frac{\left(\frac{D}{D_c}\right)^2}{\sqrt{1 - \left(\frac{D}{D_c}\right)^2}} \qquad (1)$$

$$E_b = 2\pi At \frac{r_0}{\Delta} \qquad (2)$$

Here $t$ is the total FM film thickness (assuming identical materials for the two FM layers). Two physical quantities govern the AF skyrmion size and stability: (i) the ratio $D/D_c$ where $D_c = 4\sqrt{AK_{eff}}/\pi$ is the critical DMI constant for which the single-domain state is no longer stable (negative DW energy), (ii) the DW width $\Delta = \sqrt{A/K_{eff}}$. Equation (1) shows that the radius $r_0$ increases with $\Delta$ and $D/D_c$, and diverges as

$D/D_c$ approaches 1. To reduce the DW energy and favour the nucleation of skyrmions, the SAF stack was adjusted to be close to the spin reorientation transition, i.e., with low $K_{eff}$. This low $K_{eff}$ leads to a large DW width $\Delta = 20$ nm as well as a ratio $D/D_c = 0.94$ close to 1. Thus, the large skyrmion size in our SAF multilayer can be accounted for the low DW energy and the large DW width resulting from the low PMA.

In ultra-thin films, the PMA and DMI are of interfacial origin and scale as $1/t$, with $t$ the magnetic thickness. This provides a way to tune the skyrmion size and stability by varying the film thickness. This is confirmed by micromagnetic simulations of the skyrmion diameter and stability energy $E_b$ as a function of the Co film thickness (see Fig. 3a, b, red dots). The skyrmion diameter increases rapidly over a narrow range of film thickness around 1.39 nm, corresponding to our experiments. A similar trend occurs for the skyrmion stability energy. This is explained by the approximately linear relation between the skyrmion size and its energy (see inset of Fig. 3b): the larger the skyrmion, the more stable it is. Above 1.4 nm, no stable skyrmion state is obtained in the simulations. The dependence of the skyrmion diameter and stability energy on the film thickness predicted by Eq. (1) is plotted in Fig. 3a, b (black line) and shows a good agreement with the simulations (red dots). Eq. (1) allows us to explain the divergence of the skyrmion diameter and stability energy when approaching the threshold around 1.4 nm. In this thickness range, $K_{eff}$ approaches zero (spin reorientation transition) when the thickness increases. This leads to a divergence of the DW width $\Delta$ (see Fig. 3c) and to $D/D_c$ approaching 1, i.e., the DW energy tending toward zero. Both these effects lead to a fast increase of the skyrmion radius when increasing the film thickness.

These results show that the skyrmion diameter can be reduced down to 10 nm by tuning the thickness of the layer constituting the SAF, while maintaining a significant thermal stability (~$25k_BT$). Note that we consider here two repetitions of the SAF bilayer (4 FM layers), but the stability can be further enhanced by increasing the number of repetitions since it scales linearly with the magnetic thickness in the limit of ultra-thin films. Furthermore, a larger stability energy for a given skyrmion size and thickness can be achieved by considering materials with a larger exchange constant $A$, or with smaller DW width, i.e., larger PMA, as suggested by recent ab initio calculations[50].

## Skyrmion nucleation via current injection and laser excitation

In the experiments described above, the skyrmion nucleation using either the external magnetic field or current injection is stochastic and occurs on random sites in the track. However, a local and controlled

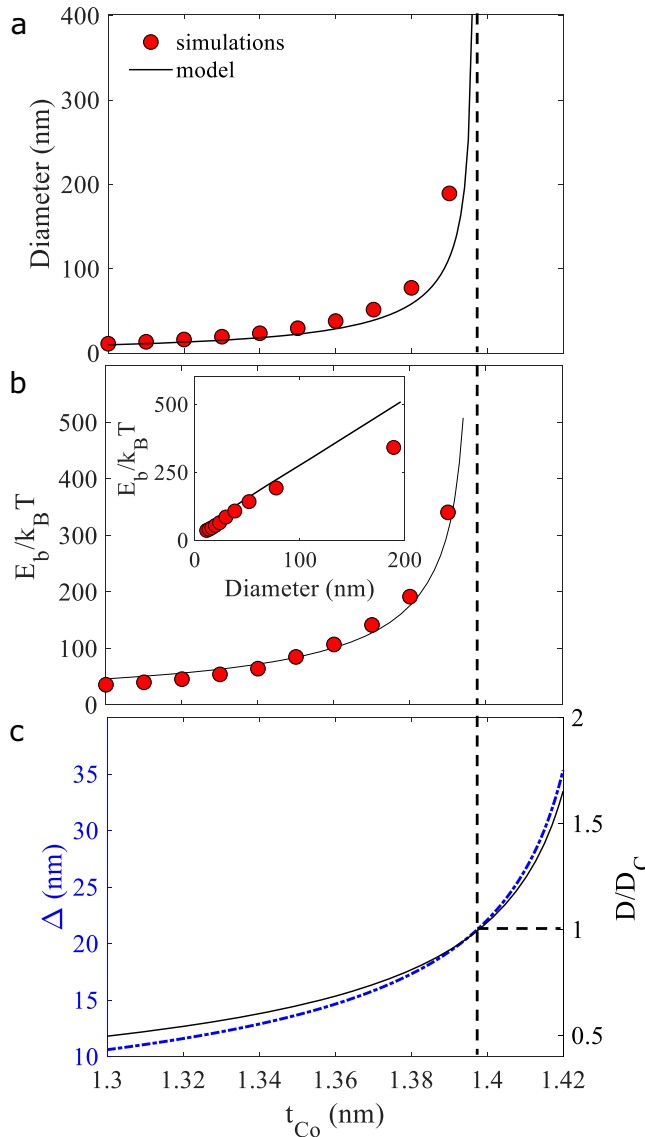

**Fig. 3 | Skyrmion size and stability vs film thickness. a** Skyrmion diameter and **b** stability energy $E_b$ (in units of $k_B T$ with $T = 300$ K) as a function of the Co film thickness $t_{Co}$ predicted by micromagnetic simulations (red dots) and the analytical model (black line). Inset: stability energy vs skyrmion diameter. **c** Domain wall width $\Delta$ and ratio of DMI to critical DMI $D/D_c$ as a function of the Co film thickness. The thickness of the Co/NiFe/Co layer is adjusted at each $t_{Co}$ such that the SAF is always compensated.

skyrmion nucleation is required for the write operation in devices. Here, we show that it can be achieved using either pulsed current injection or ultra-fast laser excitation, both at zero external field.

For the local current injection, a triangular-shaped gold injector was patterned onto a 2-μm-wide track (see STXM image in Fig. 4a): the large current density at the apex of the triangle leads locally to large heating, spin accumulation, and diverging current lines which can trigger the skyrmion nucleation under the tip[7,17,51]. Here, a SAF multilayer with a larger PMA was used so that the tracks are uniformly magnetised at remanence (see Methods, SAF2); the SAF character of the spin textures was confirmed by XMCD images acquired at different absorption edges (see Supplementary Information, S1.2.3). Starting from this single-domain state at zero field, the injection of a 5 ns current pulse in the track leads to the nucleation of a magnetic skyrmion under the tip (see Fig. 4b). The skyrmions can then be annihilated by the subsequent injection of a

current pulse with opposite polarity (Fig. 4c). This process is reproducible: the skyrmion can be nucleated and annihilated back and forth by the successive injection of current pulses with opposite polarities (Fig. 4d–f) and the nucleated skyrmion size remains fairly constant, around 420 nm. Furthermore, the process is found to be independent of the initial direction of the magnetisation. Micromagnetic simulations using experimental parameters show that the skyrmion nucleation/annihilation process can be explained by the inhomogeneous current-induced spin-orbit torque under the metallic tip (see Supplementary Information, S1.2.4).

While local current injection allows low-power and scalable write operations, the SAF skyrmion nucleation can also be achieved using ultra-fast laser pulse illumination. We show in Fig. 5a an XMCD-PEEM image of a compensated SAF multilayer [Pt(0.75)/Co(1.49)/Ru(0.85)]$_6$ (see Methods, SAF3) at zero field exhibiting uniform magnetisation. The thickness of the FM layer was adjusted close to the spin reorientation transition in order to reduce the nucleation energy barrier, although the magnetic layers remain perpendicularly magnetised with large magnetic domains (see Supplementary Information, S1.3). The illumination of the sample by ultra-fast (80 fs) laser pulses (beam size $(4.3 \pm 1) \times (6.3 \pm 1)$ μm$^2$ and repetition rate 1.25 MHz) leads to the nucleation of several isolated SAF skyrmions at the laser spot position (Fig. 5b). Note that despite their vanishing magnetic moment, the SAF skyrmions are detectable with large magnetic contrast using XMCD-PEEM. This is explained by the surface sensitivity of the technique, which results in a larger contribution to the magnetic contrast of the FM layers closer to the surface. The nucleation of worm domains and isolated skyrmions on a larger area by laser pulse trains (see (Fig. 5c) as well as single 80 fs laser pulses with larger fluence is also observed (see Supplementary Information, S1.3.3).

The nucleation of magnetic skyrmions and skyrmion lattices by ultra-fast laser pulses was demonstrated recently in ultra-thin films[52–54] and can be explained by the laser-induced heating which allows us to overcome the nucleation energy barrier. To confirm this picture, we carried out numerical calculations of the temperature during the laser pulse illumination using the 3-temperature model[55] (see Supplementary Information, S1.3.4). The simulations show that the laser illumination leads to a fast (400 fs) increase of the spin temperature up to 1270 K, well above the Curie temperature of Pt/Co multilayers (550-650 K)[56,57] and close to the one of bulk Co (1388 K). This time scale is in line with time-resolved measurement of the demagnetisation process by ultra-fast laser pulse excitation in Pt/Co multilayers[57,58]. To explain the SAF skyrmion nucleation from the demagnetised state, we carried out micromagnetic simulations using experimentally derived parameters. To mimic the demagnetisation process, we considered a magnetisation distribution with random orientation as initial state. The simulations show that it takes a few hundreds of ps for isolated skyrmions to get nucleated, their size reaching equilibrium over a time scale of the order of 10 ns (see Supplementary Information, S1.3.5). After relaxation, SAF skyrmions are eventually stabilised (see Fig. 5d). Note that, in contrast to FM skyrmions, no external magnetic field is needed to nucleate SAF skyrmions from the uniformly magnetised state using ultra-fast laser pulses.

In conclusion, we have demonstrated that skyrmions can be stabilised at room temperature and zero external magnetic field in compensated synthetic antiferromagnets. Micromagnetic simulations and X-ray microscopy confirm the left-handed Néel character of the SAF spin textures. An analytical model allows the identification of the physical parameters controlling the skyrmion size and stability, which can be easily tuned by adjusting the thickness of the different constituent layers. Since the skyrmion nucleation in SAFs is challenging due to their vanishing magnetic moment, we then studied the potential of local excitation. First, by designing current injectors, we showed the current-induced controlled and reversible nucleation and

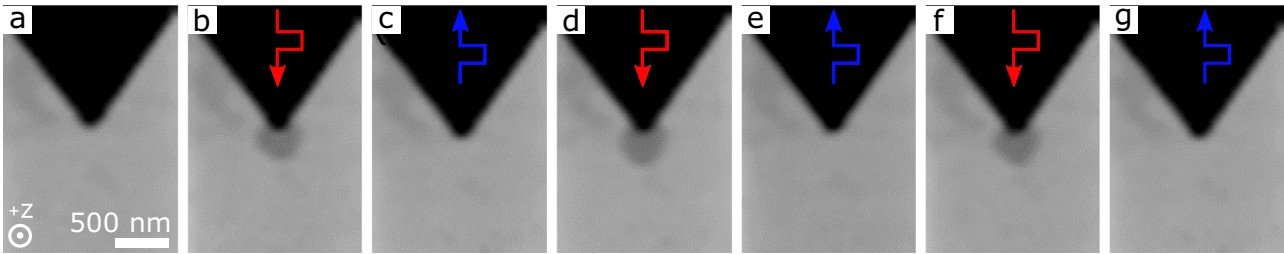

**Fig. 4 | Current-induced nucleation/annihilation of SAF skyrmions.**
**a**–**g** Sequence of STXM images acquired at the Co edge. Before each acquisition, a single 5 ns current pulse with density $J = 6.2 \times 10^{11}\,A\,m^{-2}$ is injected in the direction indicated by the red and blue arrows. The layer composition is [Pt/FM1/Ru/Pt/FM2/Ru]$_{12}$ with FM1 = Co(0.2)/NiFe(0.95)/Co(0.2) and FM2 = Co(0.9) (thicknesses in nanometres). No external magnetic field is applied.

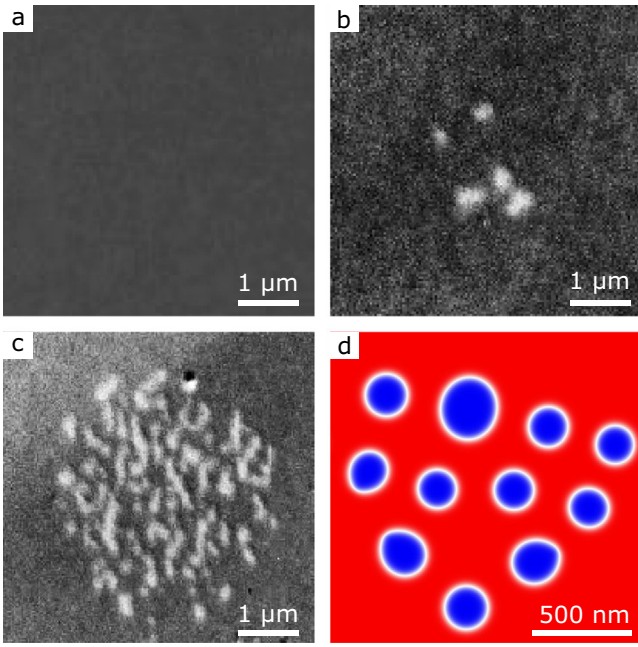

**Fig. 5 | Nucleation of SAF skyrmions by ultra-fast laser pulses. a, b** XMCD-PEEM images (Co $L_3$ edge) of a Pt(2.25)/[Pt(0.75)/Co(1.49)/Ru(0.85)]$_6$ (thicknesses in nanometres) SAF multilayer (**a**) before and (**b**) during the illumination of the sample by ultra-fast (80 fs) laser pulse excitation with repetition rate of 1.25 MHz and a fluence of $10 \pm 3.5\,mJ\,cm^{-2}$ in the centre of the beam (**c**) XMCD-PEEM image of worm domains and skyrmions after the continuous illumination (several minutes) by ultra-fast laser pulses with a fluence of $7.0 \pm 2.5\,mJ\,cm^{-2}$. The experiments were carried out at zero field. **d** Micromagnetic simulation using experimental parameters. Magnetisation distribution ($m_z$ in colour scale (cf Fig. **2a**) obtained after relaxation from a random initial state, mimicking a fully demagnetised state resulting from the laser excitation. Scale bars in **a**, **b**, **c**, **d** are 1 μm and 500 nm, respectively.

slightly different, the samples all have very similar magnetic properties (see Supplementary Information, S1) and bear the same underlying physics, making the current- and laser-induced nucleation mechanisms applicable to any of the three systems. The SAF multilayers are as follows: (i) SAF1, composed of Pt(2.5)/[Pt(0.5)/Co(1.35)/Ru(0.85)/ Pt(0.5)/Co(0.3)/Ni$_{80}$Fe$_{20}$(1.45)/Co(0.3)/Ru(0.85)]$_2$/Pt(2) (thicknesses in nanometres), was optimised for the observation of SAF skyrmions by STXM (Fig. 1), made possible by the presence of different FM materials in the SAF; (ii) SAF2, composed of Ta(3)/Pt(3)/[Pt(0.5)/Co(0.2)/ Ni$_{80}$Fe$_{20}$(0.95)/Co(0.2)/Ru(0.85)/Pt(0.5)/Co(1.35)/Ru(0.85)]$_{12}$/Pt(2), was optimised with a slightly larger PMA, so as to facilitate the observation of the current-induced skyrmion nucleation and annihilation by starting from a uniformly magnetised film (Fig. 4); (iii) SAF3, composed of Ta(3)/Pt(2.25)/[Pt(0.75)/Co(1.49)/Ru(0.85)]$_6$/Pt(1.2) was used for the laser-induced skyrmion nucleation experiments, observed by XMCD-PEEM (Fig. 5). SAF3 was kept close to the spin reorientation transition so as to minimise the skyrmion nucleation energy barrier. In addition, using surface-sensitive XMCD-PEEM allowed us to use a stack composed of only one FM material (Co), simpler than SAF1 and SAF2, that were optimised to observe the antiparallel alignement of the skyrmions in the different FM layers by STXM. SAF1 and SAF2 were deposited by magnetron sputtering on both a 100 mm Si (100) substrate and 200-nm thick 100 × 100 μm² Si$_3$N$_4$ membranes for the STXM experiments. The NiFe layer was deposited as a wedge, such that its thickness varies linearly along the 100 mm wafer between 0.76 nm and 1.55 nm. The films deposited on the membrane were patterned into 2-μm-wide tracks using electron beam lithography and ion beam etching, and then were contacted by Ti(10 nm)/Au(100 nm) pads using UV lithography. On some samples, a triangular-shaped injection tip, consisting of Ti(10 nm)/Au(100 nm), was patterned at one end of the track for the skyrmion nucleation using e-beam lithography and ion beam etching. SAF3 was deposited by magnetron sputtering on a Si wafer as well as on transparent MgO substrates for the laser excitation experiments. The FM Co layers were deposited as a wedge, with thickness varying from 1 to 1.8 nm along the 100 mm Si wafer. More details regarding the optimisation of the SAFs and their characterisation can be found in the Supplementary Information.

annihilation of skyrmions in an otherwise uniformly magnetised track. Second, we demonstrate that isolated SAF skyrmions can be nucleated from a uniformly magnetised state by 80 fs laser pulse excitation at zero magnetic field. The possibility to locally nucleate skyrmions by using current injection or ultra-fast laser excitations in compensated SAFs offers promising perspectives to design devices based on the manipulation of SAF skyrmions.

## Methods
### Sample preparation
Three compensated SAF multilayers are discussed in the main text. They were designed with slightly different compositions to obtain optimal observation conditions in each experiment, depending on the observation and nucleation methods. Although their composition was

### X-ray magnetic microscopy
Scanning transmission X-ray magnetic microscopy experiments were carried out at the Hermes beamline of the Soleil synchrotron, Saint-Aubin, France; at the Maxymus endstation at the BessyII electron storage ring operated by the Helmholtz-Zentrum Berlin für Materialien und Energie; at the PolLux (X07DA) beamline of the SLS synchrotron, Villigen, Switzerland. All experiments were carried out at room temperature. The X-ray beam was impinging the sample surface perpendicularly to the sample surface, such that the magnetic contrast is proportional to the z component of magnetisation, normal to the sample plane. The ptychographic reconstructions were carried out at the Hermes beamline of the Soleil synchrotron. For that purpose, the following procedure was used: the point detector classically used for

STXM (here a photo-multiplier tube) has been replaced by a CMOS camera[59]. The whole data set is obtained by scanning the sample with a 53% overlap of the probe. It is then reconstructed using PyNX software[60] with the Alternate Projection algorithm[61]. XMCD-PEEM imaging experiments were carried out at the CIRCE beamline[62] of the ALBA synchrotron, Barcelona, Spain, and at the UE49-PGMa beamline of the BESSYII synchrotron, Berlin, Germany. The sample surface is illuminated by the X-ray beam incident at a grazing angle of 16°. The magnetic contrast is proportional to the component of magnetisation along the X-ray beam. All experiments were carried out at room temperature. The laser-induced nucleation of SAF skyrmions were carried out at the SPEEM beamline[63]. The Gaussian-shaped laser pulses were generated by a Femtolasers Scientific XL Ti:Sapphire oscillator with a central wavelength of 800 nm, a pulse duration of 80 fs (full width at half maximum) and circular polarisation. The SAF stack was grown on a MgO substrate and the laser light is focused normal to the backside of the sample. The size of the laser beam was estimated to be around $(4.3 \pm 1) \times (6.3 \pm 1) \, \mu m^2$ as measured in previous experiments from the PEEM image of the laser beam[64].

## Micromagnetic simulations

The micromagnetic simulations were carried out using Mumax3[65]. The energy $E_b$ is defined as $E_b = E_0 - E_t$, where $E_t$ is the skyrmion energy with respect to the uniform state as obtained from the micromagnetic simulations, and $E_0 = \sum_i 8\pi A_i t_i$ is the zero radius limit of the exchange energy in the continuous limit[14,15,66,67] ($A_i$ and $t_i$ are the exchange constant and thickness of the $i^{th}$ layer respectively). More details about the simulations and magnetic parameters can be found in the Supplementary Information.

## Data availability

The raw data associated with the figures of the main text have been deposited in the Open Science Framework repository under the adress https://doi.org/10.17605/OSF.IO/R4V5M.

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

## Acknowledgements

The authors would like to thank Gregory Malinowski for his help in the calculation of the laser absorption coefficients. We thank Jordi Prat for his support at ALBA synchrotron, Helmholtz-Zentrum Berlin für Materialien und Energie for the allocation of synchrotron radiation beamtime. We acknowledge the support of the Agence Nationale de la Recherche, project ANR-17-CE24-0045 (SKYLOGIC), the support of the DARPA TEE program through grant MIPR# HR0011831554 from the DOI, the support of ALBA synchrotron through the CALIPSOplus (Grant 730872) funding, and the support of the PolLux beamline financed by the German Ministerium für Bildung und Forschung through contracts 05K16WED and 05K19WE2.

## Author contributions

R.J. and O.B. conceived and designed the experiments. R.J., G.G. and O.B. fabricated the devices. R.J., N.S., J.U.L., Q.Z., V.T.P., K.G.R., B.S., N.M., S.S., R.B., M.A.M., N.N.M., F.K., M.W., J.G., S.W., S.F., J.R., L.A., M.F. and O.B. participated in the X-ray microscopy experiments. R.J., N.S., J.U.L. and O.B. analysed the microscopy data. R.J., Q.Z. and V.T.P. performed the VSM and pMOKE characterisations. M.B., J.M.S. and H.T.N. carried out the BLS experiments. N.S., L.B.P. L.R. carried out the micromagnetic simulations and analytical modelling. O.B. and J.P.P. carried out the 3-temperature model simulation. R.J., N.S. and O.B. wrote the manuscript. All authors provided critical input to the scientific discussion and proofreading of the paper.

## Competing interests

The authors declare no competing interests.
