## [Peer Review File · Nature Communications]

Reviewers' Comments:

Reviewer #1:

Remarks to the Author:

Antiferromagnetic skyrmions are prospective for skyrmion-based spintronic memory and logic devices. In this work, the authors report the stabilization of antiferromagnetic skyrmions at zero applied magnetic field by electrical current injection or ultrafast laser pulses, in a synthetic antiferromagnet (SAF) structure. Using scanning tunneling X-ray microscopy measurements, the authors directly visualize the antiferromagnetic nature of the skyrmions and compare the obtained size and thermal stability of the skyrmions with theoretical models. The results show that it is possible to stabilize skyrmions of sizes ~ 10 nm in SAF structure with properly engineered stack structures. The results are solid and has been presented in a clear and convincing manner. Overall, I believe this paper to be of importance in the spintronics community and should be published subject to the following questions and comments:

- (1) The authors state that the observed skyrmion diameter is 204 nm with deviation of 85 nm. Roughly speaking, the size variation is 50% of the mean skyrmion diameter nucleated by the authors. Since the stabilization of isolated skyrmions are indeed of utmost importance for skyrmion-based memory and computing devices, the authors should at least comment on some strategies to reduce this large size variation while nucleation. They should also mention if this size variation is not an issue.
- (2) Page 6: "Fig. 1(e) shows a zero field STXM image of a track with large domains and an isolated skyrmion prepared after applying successively a magnetic field of 180 mT out of then in the plane of the film." The sentence needs to be revised.
- (3) The authors state that NiFe layer in their SAF structure enables them to experimentally visualize the magnetic textures in the various layers of the SAF structure. I am interested to know if the NiFe layer can also reduce the pinning potential for skyrmion dynamics.
- (4) The nucleation of skyrmions were achieved either by using current pulses or laser pulses. From the supplementary information figures S8 and S11, it appears that there might be a difference between the two. It is not clear to me if one of them is better than the other to achieve controlled nucleation. Given the results, it might not be clear to the readers as to why nucleation were done with currents and laser pulses.
- (5) Figs. 3(a), (b) shows the skyrmion diameter and thermal stability as a function of ferromagnet thickness. Is there a way to be able to tune either skyrmion diameter or thermal stability independently without affecting the other?

Reviewer #2:

Remarks to the Author:

In this paper, the authors report on the fabrication of synthetic antiferromagnetic skyrmions using the technique of combining thin compensated ferromagnetic bilayers. This could be a promising path forward in the search for antiferromagnetic skyrmions that are stable at room temperature, not susceptible to the skyrmion hall effect, and insensitive to magnetic fields. The authors use a combination of magnetic imaging techniques and micromagnetic simulations to confirm the presence of the skyrmions, their size, and the cohesiveness of the layers (for example whether each layer has the same handedness of skyrmion or not). This combination of techniques is relatively convincing, within the boundaries of our current experimental capabilities. The authors show as well that the skyrmions can be nucleated and annihilated back and forth by successive injection of current pulses of opposite polarities. This raises the question, did the authors attempt to move the skyrmions with current, and if so, could they show that there was no drift (i.e. skyrmion hall effect)? This would be an important note to add (either negative or positive confirmation).

The skyrmion diameter varies based on different factors that are an integral part of the stack formation. A figure or equation/table could be useful here to help the reader understand what are the key differences in the fabrication that result in difference sizes of the skyrmions and different energy barriers to their formation. Also, the authors should clearly state what they mean by a "pure" SAF and what exactly makes a SAF "pure".

The authors also show the nucleation of skyrmions via pulsed laser excitation. In order to understand their results in the context of ultrafast demagnetization, it would be useful to report

their laser fluences in more conventional units of fluence: mJ/cm^2 . Once the conversions are done, it appears that they are using a very large amount of pulse energy on each pulse. This is important to understand the process. It seems that the material is being heated past the Curie temperature, and then when it relaxes it forms the skyrmion state once again. This is also how the micromagnetic simulations assumed their formation, however a small calculation of the heat capacity and the maximum temperature reached by the stack given the incident fluence would be a strong addition to the manuscript and would also aid the reader to understand the probable timescales involved in the process. Are there any references for ultrafast demagnetization in these kinds of samples? MOKE? Certainly the absorption will not be uniform for an 800nm laser beam so some approximations must be made. The authors should also state the spot size for illumination in the main text. Since this is used in the micromagnetic simulations, therefore it appears to be important for the process of nucleation.

The authors make the claim that synthetic AFM skyrmions have insensitivity to magnetic fields, yet they also show that individual skyrmions can be nucleated by large magnetic fields. It would be useful to carefully state to what extent the skyrmions are insensitive to magnetic fields, since the stated value for nucleation in the paper and SI of 180mT is not actually that high. Also, providing a reference for this statement would be useful.

In general, I feel that this is a very strong manuscript with significant new scientific findings, that would be very suitable for publication in Nature Communications. The authors used an innovative new technique to achieve fabrication of a new kind of skyrmion hosting material, at room temperature, which has important technological implications. With a few small additional calculations and explanations, I think that the authors can improve the clarity of their main points, and help the reader to better understand and utilize these findings for future science.

Reviewer: Dr. Phoebe Tengdin

Skyrmions in synthetic antiferromagnets and their nucleation via
electrical current and ultrafast laser illumination, R. Juge et al.,
Nature Communication –
Response to the reviewers' comments

Reviewer #1 (Remarks to the Author):

Antiferromagnetic skyrmions are prospective for skyrmion-based spintronic memory and logic devices. In this work, the authors report the stabilization of antiferromagnetic skyrmions at zero applied magnetic field by electrical current injection or ultrafast laser pulses, in a synthetic antiferromagnet (SAF) structure. Using scanning tunneling X-ray microscopy measurements, the authors directly visualize the antiferromagnetic nature of the skyrmions and compare the obtained size and thermal stability of the skyrmions with theoretical models. The results show that it is possible to stabilize skyrmions of sizes ~ 10 nm in SAF structure with properly engineered stack structures. The results are solid and has been presented in a clear and convincing manner. Overall, I believe this paper to be of importance in the spintronics community and should be published subject to the following questions and comments:

(1) The authors state that the observed skyrmion diameter is 204 nm with deviation of 85 nm. Roughly speaking, the size variation is 50% of the mean skyrmion diameter nucleated by the authors. Since the stabilization of isolated skyrmions are indeed of utmost importance for skyrmion-based memory and computing devices, the authors should at least comment on some strategies to reduce this large size variation while nucleation. They should also mention if this size variation is not an issue.

The large size variation observed can be attributed to different factors: (i) the disorder and inhomogeneities inherent to granular films can introduce a large dispersion in the skyrmion size [ref 1, ref 2, ref 3]; (ii) The presence of both skyrmions and multidomain state at remanence in the sample of Fig1 (see Fig.1(c)), which is due to the low DW energy, favored by the low PMA and high DMI. The repulsion between the skyrmions and surrounding domain walls leads to variation of the skyrmion size, which depends on the domain pattern. To limit this dispersion, one could optimise the SAF to exhibit a uniform magnetisation at zero field and use a reproducible nucleation method (e.g. local current injection as discussed in the paper), as well as confine the skyrmions in narrower tracks to prevent their lateral expansion (thus cutting off the large-diameter region of the histogram). A sentence was added to the manuscript on this matter on page 7.

(2) Page 6: "Fig. 1(e) shows a zero field STXM image of a track with large domains and an isolated skyrmion prepared after applying successively a magnetic field of 180 mT out of then in the plane of the film." The sentence needs to be revised.

The sentence was changed to 'Fig. 1(e) shows a zero field image of a track with stripe-like domains and an isolated skyrmion. It was obtained after applying successively an in-plane and an out-of-plane magnetic field of 180 mT.'

(3) The authors state that NiFe layer in their SAF structure enables them to experimentally visualize the magnetic textures in the various layers of the SAF structure. I am interested to know if the NiFe layer can also reduce the pinning potential for skyrmion dynamics.

To our knowledge, there is no evidence that NiFe, being polycrystalline, can lower the pinning of skyrmions. Nevertheless, one foreseen advantage of permalloy, in addition to allowing the visualisation of skyrmions with vanishing magnetic moment in SAFs, resides in its lower magnetisation as compared to Co or CoFeB, which is expected to enhance the SOT efficiency in driving magnetic skyrmions [ref 4].

(4) The nucleation of skyrmions were achieved either by using current pulses or laser pulses. From the supplementary information figures S8 and S11, it appears that there might be a difference between the two. It is not clear to me if one of them is better than the other to achieve controlled nucleation. Given the results, it might not be clear to the readers as to why nucleation were done with currents and laser pulses.

Both methods can be used to nucleate locally skyrmions in SAF. The physics behind the nucleation is indeed different and both methods have their own advantages. The current induced nucleation can be explained by the inhomogeneous spin orbit torque at the apex of the tip, although thermal activation may also play a role. One advantage of the local current induced injection is that single skyrmion can be nucleated locally using small contact, its low power, its integrability in electronic device, as well as scalability. Laser induced nucleation relies on the fast heating of the spin system which allows to overcome the topological energy barrier for nucleation. One advantage is that it is triggered by ultrafast excitation, down to 100 fs (ref 49-51), which may be exploited for applications requiring fast write operations. In addition, recent experiments (ref 51) showed that the all optical skyrmion nucleation is an intrinsic process with homogeneous spatial distribution of the nucleation probability, in contrast to spin orbit torque nucleation where the skyrmions were found to nucleate preferentially on extrinsic lateral inhomogeneities. To outline this to the reader, we add the following sentence in the main text.

While local current injection would allow low power and scalable write operations, faster excitation can be achieved laser pulse excitations.

(5) Figs. 3(a), (b) shows the skyrmion diameter and thermal stability as a function of the ferromagnet thickness. Is there a way to be able to tune either skyrmion diameter or thermal stability independently without affecting the other?

Indeed, skyrmion diameter or thermal stability can be tuned independently. From Eq.2 of the manuscript, $\frac{dE}{dr} = -\frac{2\pi A t}{\Delta}$. Thus larger energy stability for a given skyrmion diameter can be achieved by increasing the exchange constant A, or decreasing Δ . To make this clearer, we add the the following sentence in the text.

“ Furthermore, a larger stability energy for a given skyrmion size and thickness could be achieved by considering materials with a larger exchange constant A, or with smaller domain wall width, i.e larger PMA, a result in agreement with recent atomistic spin dynamics calculations [50].”

Reviewer #2 (Remarks to the Author):

In this paper, the authors report on the fabrication of synthetic antiferromagnetic skyrmions using the technique of combining thin compensated ferromagnetic bilayers. This could be a promising path forward in the search for antiferromagnetic skyrmions that are stable at room temperature, not susceptible to the skyrmion hall effect, and insensitive to magnetic fields. The authors use a combination of magnetic imaging techniques and micromagnetic simulations to confirm the presence of the skyrmions, their size, and the cohesiveness of the layers (for example whether each layer has the same handedness of skyrmion or not). This combination of techniques is relatively convincing, within the boundaries of our current experimental capabilities. The authors show as well that the skyrmions can be nucleated and annihilated back and forth by successive injection of current pulses of opposite polarities. This raises the question, did the authors attempt to move the skyrmions with current, and if so, could they show that there was no drift (i.e. skyrmion hall effect)? This would be an important note to add (either negative or positive confirmation).

We could not move the skyrmions nucleated by local current injection as they could not be detached from the electrode. The reason is the geometry of the electrode that is deposited on top of the film and therefore leads to vertical current injection. As a result, part of the nucleated skyrmion rests under the tip and the injection of additional current pulses after its nucleation rather leads to its expansion or the nucleation of another skyrmion inside the first one, forming a skyrmionium-like spin texture (see Fig. S8 in the Supplementary Materials). An optimised injector geometry would allow for the nucleated SAF skyrmions to be moved by separating the nucleation current from the motion current, for example by isolating the injector from the magnetic film as proposed by Finizio *et al.* [ref 5].

The skyrmion diameter varies based on different factors that are an integral part of the stack formation. A figure or equation/table could be useful here to help the reader understand what are the key differences in the fabrication that result in difference sizes of the skyrmions and different energy barriers to their formation.

We would like to thank the referee for her comment. We realized that in the present version of the paper, the dependence of the skyrmion size and stability on the material parameters and the relevant energy was not clearly explained. Eq1 and Eq2 are keys to explain this dependence, but this not clearly outlined in the paper. As a result, we reorganized this part of the paper in order to make this clearer.

We had the following paragraph in the revised manuscript

“ We now discuss the large skyrmion size observed experimentally.....domain wall width resulting from the perpendicular anisotropy”.

Also, the authors should clearly state what they mean by a “pure” SAF and what exactly makes a SAF “pure”.

We agree with the referee that this not clear. We remove this term.

The authors also show the nucleation of skyrmions via pulsed laser excitation. In order to understand their results in the context of ultrafast demagnetization, it would be useful to report their laser fluences in more conventional units of fluence: mJ/cm². Once the conversions are done, it appears that they are using a very large amount of pulse energy on each pulse. This is important to understand the process. It seems that the material is being heated past the Curie temperature, and then when it relaxes it forms the skyrmion state once again. This is also how the micromagnetic simulations assumed their formation, however a small calculation of the heat capacity and the maximum temperature reached by the stack given the incident fluence would be a strong addition to the manuscript and would also aid the reader to understand the probable timescales involved in the process. Are there any references for ultrafast demagnetization in these kinds of samples? MOKE? Certainly the absorption will not be uniform for an 800nm laser beam so some approximations must be made. The authors should also state the spot size for illumination in the main text. Since this is used in the micromagnetic simulations, therefore it appears to be important for the process of nucleation.

We would like to thank the referee for this comment which allows us to improve the manuscript. The value of the fluence was converted in conventional units. We also realized there were some errors in the evaluation of the fluence, and this was corrected. Using this value, we performed simulation of the temperature during the pulse using the three temperature model as well as the two temperature + magnetization model. Typical parameters for the spin, lattice, electronic specific heats and electron-lattice, electron-spin and spin-lattice interaction constants of Co/Pt stacks were used. The absorption within the different layers was estimated from the transfer matrix model using a TMM module in python (<https://pythonhosted.org/tmm/tmm.html>). The temperatures at the center of the Gaussian beam are shown in figure1~c. The laser illumination leads to a fast increase of the spin temperature up to a maximum of 1268 K at 379 fs. This value is high compared to the Curie temperature expected in Co/Pt ultrathin films (550-650 K) and close to the Curie temperature of bulk Co (1388 K). The time scale is in line with time resolved measurement of the demagnetization process by ultrafast laser pulse excitation in Co/Pt multilayers^{1,2}. The longer time scale remagnetization process is not included in the model and depend on the thermal lattice diffusion. Time resolved experiments show that it is typically in the tens of ps range for Co/Pt multilayers on Si substrate for similar fluence^{2,3}. For instance, a remagnetization of (Co/Pt) multilayers of around 10 ps is measured by Shim² after excitation by a 800 nm wavelength ultrafast laser excitation. The spot size for illumination is now included in the main text.

A description of this simulations has been added in the supplementary information as well as in the main text.

Figure 1 Three temperature model of laser excitation. Evolution of the electron, lattice and spin temperature following a 80 femtosecond laser pulse irradiation.

The authors make the claim that synthetic AFM skyrmions have insensitivity to magnetic fields, yet they also show that individual skyrmions can be nucleated by large magnetic fields. It would be useful to carefully state to what extent the skyrmions are insensitive to magnetic fields, since the stated value for nucleation in the paper and SI of 180mT is not actually that high. Also, providing a reference for this statement would be useful.

We agree with the reviewer that, strictly speaking, skyrmions in SAFs are not insensitive to external fields but rather to moderate fields below the RKKY field limit (as we mentioned in the text). In the experiments presented in Fig. 1, the 180 mT magnetic field applied happens to be around that limit (see hysteresis loop in Fig. 1(b)). In addition, SAF domains and skyrmions can hardly be completely compensated due to uncompensated moments in the AF-coupled domain walls that act as nucleation centres for magnetisation reversal in SAFs [ref 7]. Nonetheless, below the RKKY field limit, SAF skyrmions are expected to remain largely unaffected by external fields [ref 7, ref 8]. A sentence was added to the manuscript on this matter on page 6.

In general, I feel that this is a very strong manuscript with significant new scientific findings, that would be very suitable for publication in Nature Communications. The authors used an innovative new technique to achieve fabrication of a new kind of skyrmion hosting material, at room temperature, which has important technological implications. With a few small additional calculations and explanations, I think that the authors can improve the clarity of their main points, and help the reader to better understand and utilize these findings for future science.

Reviewer: Dr. Phoebe Tengdin

OTHER MODIFICATIONS TO THE MANUSCRIPT

- The abstract was modified to fit into the 150-word limit.
- Added a 'Data availability' section.
- Added an 'Author contributions' section.
- Added a 'Competing interests' section.
- Changed reference style as per formatting instructions.
- Johan Pelloux-Prayer was added as a co-author of the paper. He performed the temperature calculation during the laser excitation which were asked by Referee 2.

Reference

1. Kuiper, K. C. *et al.* Spin-orbit enhanced demagnetization rate in Co/Pt-multilayers. *Applied Physics Letters* **105**, 202402 (2014).
2. Shim, J.-H. *et al.* Ultrafast dynamics of exchange stiffness in Co/Pt multilayer. *Commun Phys* **3**, 1–8 (2020).
3. Vaskivskiy, I. *et al.* Element-Specific Magnetization Dynamics in Co–Pt Alloys Induced by Strong Optical Excitation. *J. Phys. Chem. C* **125**, 11714–11721 (2021).

Reviewers' Comments:

Reviewer #1:

Remarks to the Author:

The authors have satisfactorily answered to all the concerns raised by the reviewers and there are no more questions from my side. The manuscript is now suitable for publication in Nature Communications.

Reviewer #2:

Remarks to the Author:

This paper represents an important advance in experimental demonstration of antiferromagnetic skyrmions. It appears that the authors have taken all considerations from the reviewers seriously, and improved the manuscript in several important way. They have clarified the relationship between size and stability, as well as performed additional simulations regarding the laser driven nucleation process.

I would like to point out however that since the authors do not show any proof that the formation dynamics by femtosecond laser take place on ultrafast timescales, and in fact their simulations do not address this question either, they should change the statement:

"While local current injection allows low power and scalable write operations, faster excitations can be achieved using ultrafast laser pulse illuminations "

To "faster excitations could be achieved using ultrafast..."

I suggest that the authors make an estimate of the timescale for the relaxation induced formation dynamics of the skyrmions using the micromagnetic simulations that they have performed. This process is shown in Figure S13 of the SI but no time labels are given on the plots. If an estimate cannot be made from simulations at least they could give one by consulting the literature here.

If the above changes are made, I believe that all concerns initially raised by both reviewers have been sufficiently answered and would recommend this article for publication in Nature Communications. Although an exact number on the formation timescales is certainly beyond the scope of this work, I believe that this study will catalyze future work that could quantify the precise timescale of skyrmion formation, etc. and will become an important work in the field of skyrmion based spintronics.

Response to Referees

Reviewer #1 (Remarks to the Author):

The authors have satisfactorily answered to all the concerns raised by the reviewers and there are no more questions from my side. The manuscript is now suitable for publication in Nature Communications.

Reviewer #2 (Remarks to the Author):

This paper represents an important advance in experimental demonstration of antiferromagnetic skyrmions. It appears that the authors have taken all considerations from the reviewers seriously, and improved the manuscript in several important way. The have clarified the relationship between size and stability, as well as performed additional simulations regarding the laser driven nucleation process.

I would like to point out however that since the authors do not show any proof that the formation dynamics by femtosecond laser take place on ultrafast timescales, and in fact their simulations do not address this question either, they should change the statement: "While local current injection allows low power and scalable write operations, faster excitations can be achieved using ultrafast laser pulse illuminations"

To "faster excitations could be achieved using ultrafast..."

In the assertion "faster excitations can be achieved using ultrafast laser pulse illuminations", we mean that the excitation by the laser illumination pulse is 80 fs long, while the excitation by the injection of a current pulse is ns long. However, to avoid confusion between excitation time and nucleation time for the reader, the sentence is replaced by

"While local current injection allows low power and scalable write operations, the SAF skyrmion nucleation can also be achieved using ultrafast laser pulse illumination"

I suggest that the authors make an estimate of the timescale for the relaxation induced formation dynamics of the skyrmions using the micromagnetic simulations that they have performed. This process is shown in Figure S13 of the SI but no time labels are given on the plots. If an estimate cannot be made from simulations at least they could give one by consulting the literature here.

The simulations of figure S13 were performed using the minimization of the micromagnetic energy using the steepest conjugate gradient method. The different images show the evolution of the magnetization configuration at different steps during the energy minimization procedure. This explains why there is no time label given on the plots. We used this procedure to decrease the computation time to converge to the equilibrium state. Indeed, the simulations include 6 layers which are AF coupled, with a large number of cells (1536x1536, cell size 1 nm). However, to address this comment and estimate the time relaxation, we eventually solve the LLG equation to obtain the time evolution

of the magnetic configuration after the demagnetization. This is shown in the new Figure S13 reproduced below

Figure 1 Micromagnetic simulations - Laser-induced SAF skyrmion nucleation. Evolution of the magnetization at different times (from a to f), starting from an initial circular demagnetized state mimicking the laser-induced demagnetization (a).. The simulation window size is 1536x1536 nm². The cell size is 1 nm.

These results are commented as follows in the supplementary

“ Figure S13 shows the magnetization configuration at different times during the relaxation as obtained by solving the LLG equation (damping parameter of 0.3). At short time scales (Figure S13(b), $t=0.05$ ns), small up and down domains connected by in-plane magnetized regions appear, which can be accounted for the fast relaxation of the Heisenberg exchange energy. Larger domains with perpendicular magnetization form rapidly after (Figure S13(c), $t=0.2$ ns), which can be explained by the relaxation of the magnetic anisotropy energy. Small isolated skyrmions are also observed at this stage. The domains then increase in size and appear less dense (Figure S13(d), $t=0.4$ ns). While most of the smallest skyrmions turned out to be unstable and disappeared, the larger isolated skyrmions eventually stabilize at this stage. Their size slowly converges to the equilibrium value (Figure S13(e-f), $t=1-6$ ns), which can be accounted for the relaxation of the domain wall energy. Note that the relaxation is slow after 0.4 ns and that due to the limited computation time (about 1 week), the final equilibrium state could not be reached. The latter was obtained by minimizing the micromagnetic energy using the steepest conjugate gradient method (function “minimize” in Mumax3 [10]) and is shown in Fig.5(d) of the main text. These results show that the typical time scale to relax to the equilibrium state is larger than 6 ns and likely of the order of a few tens of ns, in line with similar simulations for ferromagnetic skyrmions where a relaxation time scale of a few tens of ns was observed [24]. Finally note that although micromagnetic simulations capture most of the physics of the skyrmion formation after the laser-induced demagnetization on a longer time scale, atomistic spin dynamics calculations would be needed to describe the laser-induced demagnetization and the dynamics right after the laser pulse on the ps time scale”

Since the simulations were performed using a different initial demagnetized state generated with different random seed, the final equilibrium state was different from the one of the previous submitted version of the manuscript. Thus, we replace in Fig5d the final equilibrium state with the new one. The new figure 5 is shown below.

Figure 2 New figure 5 of the main text

The following text was added to comment on the typical time scale of relaxation.

Simulations show that it takes a few hundreds of ps for isolated skyrmions to get stabilized, their size eventually converging to the equilibrium one over a time scale of the order of ten of ns (see Supplementary Information, figure S13).

If the above changes are made, I believe that all concerns initially raised by both reviewers have been sufficiently answered and would recommend this article for publication in Nature Communications. Although an exact number on the formation timescales is certainly beyond the scope of this work, I believe that this study will catalyze future work that could quantify the precise timescale of skyrmion formation, ect . and will become an important work in the field of skyrmion based spintronics.